# Evaluation of Adjuvant Activity and Bio-Distribution of Archaeosomes Prepared Using Microfluidic Technology

**DOI:** 10.3390/pharmaceutics14112291

**Published:** 2022-10-26

**Authors:** Yimei Jia, Gerard Agbayani, Vandana Chandan, Umar Iqbal, Renu Dudani, Hui Qian, Zygmunt Jakubek, Kenneth Chan, Blair Harrison, Lise Deschatelets, Bassel Akache, Michael J. McCluskie

**Affiliations:** 1Human Health Therapeutics, National Research Council Canada, Ottawa, ON K1A0R6, Canada; 2Nanotechnology Research Centre, National Research Council Canada, Edmonton, AB T6G2M9, Canada; 3Metrology Research Centre, National Research Council Canada, Ottawa, ON K1A0R6, Canada

**Keywords:** archaeosome, vaccine adjuvant, sulfated lactosyl archaeol, bio-distribution, glycolipid, liposome, microfluidics

## Abstract

Archaeosomes, composed of sulfated lactosyl archaeol (SLA) glycolipids, have been proven to be an effective vaccine adjuvant in multiple preclinical models of infectious disease or cancer. They have classically been prepared using a thin-film hydration method with an average particle size of 100–200 nm. In this study, we developed methods to generate SLA archaeosomes at different sizes, i.e., 30 nm and 100 nm, via microfluidic mixing technology and evaluated their physicochemical characteristics, as well as adjuvant activity and in vivo biodistribution in mice. Archaeosomes, prepared using thin-film and microfluidic mixing techniques, had similar nanostructures and physicochemical characteristics, with both appearing stable during the course of this study when stored at 4 °C or 37 °C. They also demonstrated similar adjuvant activity when admixed with ovalbumin antigen and used to immunize mice, generating equivalent antigen-specific immune responses. Archaeosomes, labeled with CellVue^TM^ NIR815, had an equivalent biodistribution with both sizes, namely the highest signal at the injection site at 24 h post injection, followed by liver, spleen and inguinal lymph node. The presence of SLA archaeosomes of either size helped to retain OVA antigen (OVA-Cy5.5) longer at the injection site than unadjuvanted OVA. Overall, archaeosomes of two sizes (30 nm and 100 nm) prepared using microfluidic mixing maintained similar physicochemical properties, adjuvant activity and biodistribution of antigen, in comparison to those compared by the conventional thin film hydration method. This suggests that microfluidics based approaches could be applied to generate consistently sized archaeosomes for use as a vaccine adjuvant.

## 1. Introduction

Adjuvants are key components of subunit vaccines, inducing effective immune responses against poorly immunogenic antigens. Archaeosomes are comprised of archaea-derived total polar lipids (TPL) or semi-synthetic glycerolipids of ether-linked isoprenoid phytanyl cores with varied glyco- and amino-head groups. For many years, they have been used as a vaccine adjuvant in pre-clinical studies to promote strong humoral and cell-mediated responses to entrapped antigen [1,2]. More recently, we have shown that a simplified archaeosome adjuvanted vaccine formulation, composed of sulfated lactosyl archaeal (SLA) glycolipid admixed with protein antigens, can stimulate strong humoral and cell mediated immune responses to multiple antigens in mice, including those targeting various infectious pathogens (influenza virus, Hepatitis B virus, SARS-CoV-2, Hepatitis C virus, and *Schistomiasis mansoni*) [3,4,5,6,7] or tumor types (breast cancer and melanoma) [8,9].

Traditionally SLA archaeosomes have been produced using the thin-film hydration method whereby a thin lipid film was first generated and subsequently hydrated with an aqueous buffer [10]. This process would typically result in heterogeneous large particles (polydisperity index, PDI, >0.5) that would subsequently require extensive down-sizing through extrusion or sonication to yield a more homogenous particle profile size of 100–200 nm and PDI of ~0.1. More recently, microfluidic mixing has been used to successfully generate liposomes or nucleic acid-lipid nanoparticle complexes [11,12], although this has not previously been evaluated for archaeosomes. Microfluidic mixing technology has been shown to produce nanoliposomal products in a more consistent manner by enhancing the diffusion effect between the flows of organic and water phases [12]. These two liquid streams enter a “herring bone” geometric channel, where they undergo shear forces, electric field, and microfluidic effects during the mixing, resulting in precipitation of nanoparticles within milliseconds [12,13]. Importantly, by controlling the rate of mixing and a ratio of aqueous to lipid components, the user can generate homogenous particles of various pre-determined sizes.

In the case of adjuvants, such as SLA archaeosomes, particle size may determine how they interact with components of the immune system and hence contribute to their adjuvant activity. For example, the size of a particulate vaccine can impact their ability to transport antigens to lymph nodes (LNs) where antigen presenting cells (APC), B and T cells, are located [14]. Particulate vaccine formulations and larger pathogens (500 nm–2 µm) are generally internalized by APCs via various cellular uptake mechanisms, while smaller particles such as viruses of 20–100 nm are usually taken up by endocytosis through receptor binding and internalization [15,16]. Studies have revealed that smaller-sized particles are able to extravasate freely through intracellular spaces and reach lymph nodes more readily than larger particles. It is therefore possible that smaller particles, for example <30 nm, may not stay at the site of injection long enough to induce a suitably immunocompetent environment for the uptake of co-administered antigen [17,18]. Importantly, this differential trafficking and processing can also bias responses to a specific type of immunity. For example, Fifis et al. showed that smaller nano-beads, especially those close to virus size (30–80 nm), were more efficiently localized to DEC205^+^CD40^+^CD86^+^ dendritic cells in the draining lymph node (dLN) and were more able to promote CD8^+^ and CD4^+^ type 1 T cell responses, as compared to nano-beads that had particle sizes larger than 1000 nm [19]. Ultra-small nanoparticles (20–45 nm) have been shown to be more efficiently transported by interstitial flow into lymphatic capillaries after intradermal injection resulting in longer retention in lymph nodes (up to 120 h post-injection) than particles larger than 100 nm, which require tissue-resident dendritic cells (DCs) to shuttle them to the LN [4,5]. Therefore, it was of interest to determine whether reducing the size of the archaeosomes from the customary ~100 nm to ~30 nm would alter their adjuvant activity or their distribution kinetics locally and systemically. This would also help inform us when establishing the target profile and release characteristics for SLA archaeosomes in the future.

In this study, using an orthogonal experimental design, the parameters of microfluidic mixing to prepare archaeosomes of defined sizes were optimized, particles were generated and their physicochemical characteristics were determined. Thereafter, archaeosomes were admixed with antigen and their effects on antigen-specific immune responses evaluated. Since we were using a simple admixed archaeosome/antigen formulation, it was possible that the biodistribution of each separate component could differ, hence we also investigated the effects of archaeosome particle size on biodistribution of both archaeosomes and of admixed antigen. Overall, the objective of the current work was to develop a microfluidic-based method to prepare SLA archaeosomes of different particle sizes (30 vs. 100 nm) and to evaluate their activity as vaccine adjuvants.

## 2. Materials and Methods

### 2.1. Materials and Archaeosomes Preparation by Thin-Film Hydration Method

Sulfated lactosyl archaeol (SLA; 6′-sulfate-β-D-Galp-(1,4)-β-D-Glcp-(1,1)-archaeol) was synthesized, as described previously [20]. Cy5.5-OVA conjugate was purchased from Nanocs (New York, NY, USA), while CellVue^TM^ NIR815 was purchased from Thermofisher Scientific (Waltham, MA, USA).

Thin-film hydration archaeosomes were prepared, as previously described [10]. In brief, SLA lipid that was dissolved in chloroform/methanol underwent solvent removal under N_2_ gas with mild heating to form a thin lipid film. Dried lipids were then hydrated in Milli-Q water for more than 12 h to form a uniform lipid suspension. Next, sonication was applied at 40 °C in an ultrasonic water bath (Fisher Scientific, Ottawa, ON, Canada) for up to an hour until the desired particle size (~100 nm) was obtained. After that, 10 × PBS (Millipore Sigma Canada, Oakville, Ontario) was added and archaeosomes stored at 2–8 °C until used. Note that we have previously demonstrated that archaeosomes stored under similar conditions were stable up to 6 months of testing, which was well beyond the storage time of ~1 month in the current study [21]. On the day of immunization, antigen solution (ovalbumin; OVA; type VI, Sigma-Aldrich, St. Louis, MO, USA) was added to the pre-formed archaeosomes to reach a desired final concentration.

### 2.2. Orthogonal Array Experimental Design on Preparation of Archaeosomes by Microfluidic Mixing

To identify optimal parameters to generate SLA archaeosomes by microfluidic mixing on Nanoassemblr^TM^ instrument (Nanoassemblr^TM^ Ignite, Precision Nanosystem Inc. Vancouver, BC, Canada), an orthogonal array experiment design was utilized. Three operation parameters, i.e., organic solvents, flow rate ratio (FRR; aqueous: lipid) and total flow rate (TFR; mL/min), were assigned in the following orthogonal experiments (Table 1) with a fixed SLA concentration of 10 mg/mL. The lipid concentration for microfluidic preparation was based on the instrument manufacturer’s recommendation. Archaeosomes, having a target size of 30 nm (in theory being able to flow though lymphatic capillaries to dLN) and 100 nm (which was the standard particle size for archaeosomes generated by thin-film hydration method), were prepared according to the orthogonal array design shown in Table 1. Three individual measurements were conducted to evaluate the resulting particles for particle size, zeta potential and polydispersity index (PDI). In addition, the stability of produced particles was examined following storage for three weeks at 4 °C or 37 °C to cover the period between prime and boost immunizations. As this was a pilot study, design-of-experiment (DoE) approach was used to simultaneously screen initial parameters and to identify the conditions capable of generating SLA archaeosomes with the desired profile [22].

Archaeosomes were prepared by mixing lipid stock solutions (10 mg/mL) in selected co-solvent organic systems, i.e., ethanol/methanol, ethanol/acetone and ethanol/DMSO, with aqueous PBS buffer (pH 7.4) using a microfluidic herringbone micromixer, Nanoassemblr^TM^. Desired volumes of the lipids and the buffer were simultaneously injected in the microfluidic micromixer using a dual-syringe pump (model S200, KD Scientific, Holliston, MA, USA), which push the solutions through the micromixer at defined flow rates, i.e., 8, 12, or 18 mL/min. The mixed material was diluted into 30 × volume of 25 mM PBS buffer after collection, thus reducing the ethanol content to <20% *v*/*v*. Then liposomes were purified from the ethanol-containing buffer with an Amicon Ultra, 30,000 molecular weight cut-off (MWCO) concentrator (Millipore, Billerica, MA, USA). The purification process was performed five times to ensure all residual solvents were below an acceptable daily exposure level defined by ICH guidance Q3C. This was confirmed by solvent analysis using gas chromatography.

### 2.3. Archaeosome Characterization

SLA archaeosomes prepared by microfluidic mixing using Nanoassemblr^TM^ instrument were diluted to ~0.5 mg/mL in PBS and characterized for their average size (Z-average diameter, Z-avg), PDI, and zeta potential using a Zetasizer NanoZS (Malvern Instruments, Malvern, UK). Archaeosomes preparation for cryo-transmission electron microscopy (TEM) was performed, as previously described [10,23]. All TEM images were collected in JEOL 2200FS TEM with a 200 kV accelerating voltage. The lipid concentration in the fully prepared SLA archaeosome solutions was determined using the dry weight method, as previously described [10].

### 2.4. Animals

Old female C57BL/6 or albino C57BL/6 mice aged 6–8 weeks (Charles River Laboratories, Saint-Constant, QC, Canada) were used for immunogenicity and biodistribution studies, respectively. Mice were housed at the animal facility of the National Research Council Canada (NRC) in compliance with the guidelines of the Canadian Council on Animal Care. All animal use protocols were approved by the NRC Animal Care Committee (animal use protocol #2020.10). Animals were monitored for adverse clinical signs immediately following vaccination and routinely throughout the course of the study.

### 2.5. Vaccine Immunization

Mice (n = 5 per group) were immunized by intramuscular (IM) injection of 50 µL of 10 µg OVA alone or OVA admixed with SLA archaeosomes into the left tibialis anterior (TA) muscle on Days 0 and 21. Blood was collected on Days 20 and 28 and recovered serum was used for the quantification of antigen specific IgG antibody levels. On Day 27, carboxyfluorescein succinimidyl ester (CFSE)-stained target cells diluted in Hank’s balanced salt solution (HBSS; GE Life Sciences, Chicago, IL, USA) to a final volume of 200 μL were injected into the retro-orbital plexus to assess antigen-specific in vivo cytolytic killing. On Day 28, spleens were collected for the elucidation of cellular immune responses by IFN-γ ELISpot and/or in vivo cytolytic activity assay.

### 2.6. Anti-OVA Antibody ELISA

Anti-OVA antibodies (Ab) titers in mouse serum were quantified by ELISA, as previously described [3,24]. Briefly, 96-well high-binding ELISA plates were coated with OVA protein (100 µL/well of 10 µg/mL protein in PBS) overnight. After washing, the plates were blocked with fetal bovine serum in PBS. After additional washing, serial diluted samples were added in 100 µL volumes and incubated for 1 h at 37 °C. The plates were washed five times with PBS/0.05% Tween 20 and then a 100 µL of goat anti-mouse IgG-HRP was added and incubated for 1 h at 37 °C. The substrate o-phenylenediamine dihydrochloride was added to the plate after washing. Plates were developed for 30 min at RT in the dark prior to the addition of H_2_SO_4_ to stop the chemical reaction. Titers for IgG in serum were defined as the dilution that resulted in an absorbance value (OD 450) of 0.2 and were calculated using XLfit software (ID Business Solutions, Guildford, UK).

### 2.7. ELISpot Assay

Enumeration of antigen-specific IFN-γ secreting cells was done by ELISpot assay, as described before [3,5,10,25]. Briefly, spleen cells (4 × 10^5^/well) were added to ELISpot plates coated with an anti-IFN-γ antibody (Mabtech Inc., Cincinnati, OH, USA), followed by incubation in the presence of appropriate antigen-specific stimulant at a concentration of 2 µg/mL for 20 h at 37 °C, 5% CO_2_. A peptide corresponding to CD^8+^ T cell epitope OVA_257-264_: SIINFEKL (JPT Peptide Technologies GmbH, Berlin, Germany) was used as a stimulant. To measure background responses, cells were incubated in the absence of any stimulants. The plates were developed according to the manufacturer’s instructions. AEC substrate (Becton Dickenson, Franklin Lakes, NJ, USA) was added to visualize the spots. Spots were counted using an automated ELISpot plate reader (Cellular Technology LTD, Beachwood, OH, USA).

### 2.8. In Vivo Cytolytic Activity

In vivo cytolytic activity in immunized mice was evaluated, as described previously [10,26,27]. Donor spleen-cells from syngeneic mice were prepared. Cells were split into two aliquots. One aliquot was incubated in the presence of the appropriate CTL specific peptide (10 μM SIINFEKL, JPT Peptide Technologies GmbH) in R10 media. After 30 min of incubation, the aliquot that did not contain stimulation peptides was stained with a low concentration of CFSE (0.25 μM; Thermo Fisher Scientific, Waltham, MA, USA) and the second peptide-pulsed aliquot was stained with a 10-fold higher concentration of CFSE (2.5 μM). The two aliquots were mixed at 1:1 and injected (total of 20 × 10^6^ cells/mouse) into previously immunized recipient mice. At ~20 to 22 h after the donor cell transfer, spleens were collected from recipients, single cell suspensions were prepared, and cells were analyzed using flow cytometry on a BD Fortessa flow cytometer (Becton Dickenson).

### 2.9. Fluorescence Imaging

The in vivo biodistribution of OVA-Cy5.5 (10 µg) alone or OVA-Cy5.5 (10 µg) admixed with SLA archaeosomes (1 mg) were assessed in C57BL/6 albino female mice (n = 4 per group) at various time-points following a single IM administration. Animals were subjected to in vivo imaging using an IVIS Lumina III small animal imager (Perkin Elmer, Waltham, MA, USA). Animals were imaged at pre-scan, 5 min, 10 min, 1 h, 2 h, 5 h, 24 h, 48 h, 72 h and 144 h.

Separate animals/formulations were used to monitor biodistribution of the archaeosomes themselves. The liposomes were labelled with CellVueTM-NIR815 kit (Fisher Scientific Company, Ottawa, ON, Canada) based on the manufacturer’s instructions. Briefly, archaeosomes at a particle concentration of 1 × 10^7^/mL (as determined by Zetaview, Particle Metrix GmbH, Ammersee, Germany) were labelled with NIR815 dye at a concentration of 1 µM. The labeled archaeosomes were admixed with unlabeled OVA at similar doses as above and injected into the left T. A. muscle in a volume of 50 μL. The mice were imaged at 24 h, perfused with heparinized saline and then tissues (liver, spleen, lung, heart, brain, kidneys, inguinal and muscle of injection site) collected for ex vivo imaging. The mice were imaged at excitation and emission wavelengths of 750 nm/845 nm for SLA-NIR815, and 660 nm/710 nm for Cy5.5-OVA. Total fluorescence intensity data were determined from select regions of interest (ROI) using the Living Image 4.1 software (Perkin Elmer, Waltham, MA, USA).

### 2.10. Statistical Analysis

GraphPad Prism^®^ (GraphPad Software, San Diego, CA, USA) was used for statistical analysis of the data by ANOVA between three or more groups followed by post-hoc analysis using Tukey’s multiple comparison tests. Antibody titers and ELISpot counts were log-transformed prior to the statistical analysis. Differences were considered to be significant if *p* < 0.05.

## 3. Results

### 3.1. Generation and Characterization of Archaeosomes Utilizing an Orthogonal Experimental Design Approach

An orthogonal experimental design was utilized in order to identify experimental conditions of microfluidic mixing that would generate archaeosomes that demonstrated particle characteristics (Z-avg, PDI and zeta potential) similar to those generated by the thin film hydration method (i.e., 100 nm). In addition, we were interested in identifying conditions capable of generating smaller archaeosomes with a Z-avg of ~30 nm. A summary of the characteristics of the various SLA archaeosomes generated using this approach is shown in Table 2. For comparison, archaeosomes generated using the classical thin-film method typically have Z-avg of 100–120 nm, zeta potential of ~40 mV and PDI of 0.3–0.4 [10].

Bases on the results (Table 2), two experimental runs (3 and 7) were identified as having produced particles with profiles of interest. By using ethanol and a DMSO co-solvent, FRR mixing ratio of PBS:SLA at 1:1 and total flow rate at 18 mL/min, archaeosomes with Z-avg of 94.61 nm, PDI of 0.19 and Zeta potential of −41.27 mV were produced. The size and zeta potential were similar to what we typically obtain using the thin layer hydration method, with a lower PDI indicative of more homogenous size profile. In contrast, smaller archaeosomes with Z-avg of 32.84 nm, PDI of 0.29 and Zeta potential of −33.07 mV, were generated with the following conditions: Ethanol and DMSO co-solvent, FRR mixing ratio of PBS:SLA at 4:1 and total flow rate at 8 mL/min. These formulations, referred to as SLA-100 and SLA-30, respectively, were selected for further analysis in vitro and in vivo. It is worthwhile to note that other conditions could potentially generate archaeosomes with similar characteristics, but these were deemed sufficient to achieve the goals of this study. Zeta potential values ranged from −42.0 to −19.8 mV on average. Figure 1 showed TEM images of archaeosomes generated using the two condition profiles identified above. The images revealed that SLA-100 and SLA-30 were both spherical in shape, with a bilayer surface and existed either in hollow or solid (electron dense) core compartment form. Previous TEM analysis of archaeosomes generated using the thin-film method showed a similar spherical shape, but particles only had a hollow core compartment. The formation of solid nanoparticles under microfluidic mixing in the current study was likely due to anti-solvent precipitation, resulting from the oscillatory flow that comes from fluid mixing [13,28,29].

To ensure that archaeosomes would be stable over the course of vaccine administration, we monitored potential changes of their physical properties following incubation for ~3 weeks at 4 °C or 37 °C, respectively. The physical properties of both SLA-30 and SLA-100 s remained largely unchanged over the course of the study, with the values for particle size, surface charge and polydispersity remaining largely consistent (Figure 2). This was not surprising as we have previously seen good long-term stability with archaeosomes prepared using conventional methods [21].

### 3.2. Induction of Antigen-Specific Antibody Responses by SLA Archaeosomes

Next, we compared the adjuvant activity of SLA-30 and SLA-100 to SLA generated using thin film hydration method (SLA-Control) in vivo. Following immunization of mice on days 0 and 21 with OVA alone or OVA adjuvanted with 1 mg of SLA-30, SLA-100 or SLA-Control, antigen-specific IgG responses were assessed in the serum of the immunized mice on Day 20 and 28 (7 days post 2nd dose).

Following one or two vaccine doses, similar anti-OVA IgG titers were measured in animals and in mice immunized with OVA adjuvanted with each of the three archaeosome formulations (Figure 3). Importantly, the inclusion of any of the SLA formulations with OVA significantly enhanced anti-OVA antibody IgG titers, as compared to OVA alone after either one or two immunizations (**** *p* < 0.0001). No statistically significant differences were observed in the anti-OVA IgG titers between the three different SLA adjuvanted groups.

### 3.3. Induction of Antigen-Specific Cell Mediated Immunity by SLA Archaeosomes

On Day 28, 7 days post 2nd vaccine dose, cellular responses were also assessed in the splenocytes of the immunized mice by in vivo cytolytic activity analysis. While background levels of activity (as measured by % killing of SIINFEKL-labeled cells) were measured in animals immunized with OVA alone, significantly higher killing (>50%) was observed in mice immunized with OVA adjuvanted with each of the three SLA formulations (Figure 4). No statistically significant differences were observed in the anti-OVA cytolytic activity between the three different SLA adjuvanted groups.

ELISpot was also used to enumerate the number of Ag-specific CD8 (OVA257-264-specific) T cells in the splenocytes of mice following immunization with the various OVA-containing vaccine formulations above. Mice immunized with OVA formulations containing SLA- archaeosomes showed greater numbers (*p* < 0.0001) of OVA-specific CD8+ T cells than in mice immunized with OVA alone; an average of 80–150 IFNγ^+^ SIINFEKL-specific spot-forming cells (SFC)/10^6^ splenocytes were observed, as compared to 0 SFC/10^6^ splenocytes with OVA alone (Figure 5). No statistically significant differences were observed in the number of Ag-specific CD8 (OVA257-264-specific) T cells between the three different SLA adjuvanted groups.

### 3.4. Bio-Distribution of Archaeosomes and Effect on Antigen Retention

The effect of particle sizes of SLA archaeosomes, i.e., 30 nm vs. 100 nm, on their distribution in mouse were quantified ex-vivo based on fluorescence intensity of NIR815. At 24 h post injection, whole body in vivo imaging revealed peak fluorescent signals at the injection site. Ex vivo imaging at 24 h post injection revealed fluorescence also in liver, spleen and inguinal lymph node for both SLA-30 and SLA-100 (Figure 6). There were no significant differences in tissue organ distribution between SLA-30 and SLA-100.

The effect of particle size (SLA-30 vs. SLA-100) on antigen retention following IM injection was measured at various time-points following the injection of Cy5.5-OVA alone, or in the presence of SLA archaeosomes (representative images shown in Figure 7). With the addition of archaeosomes of either size, labeled OVA was retained at the injection site at a higher level for all time-points beyond 5 h when compared to OVA alone (Figure 7).

## 4. Discussion

We have developed semi-synthetic SLA archaeosomes as a proprietary vaccine adjuvant technology and have demonstrated their efficacy in multiple preclinical animal models. We have recently demonstrated that antigen does not need to be entrapped within archaeosomes, but can be simply admixed, greatly increasing the flexibility of this vaccine adjuvant [10]. SLA archaeosomes have traditionally been made by the thin film hydration method with the size reduced to <200 nm during sterile filtration. Herein, we evaluated the use of microfluidics-based technologies in an alternative production process for these liposomes.

While the thin film hydration method has been successfully used for many years with liposome-based formulations, there are some limitations associated with it. For example, nanoparticle size distributions are often polydisperse, reproducibility from batch to batch can be difficult, there may be limited processing capacity, and peroxidative damage to the lipids may occur due to the local overheating during the process [30,31]. Microfluidic mixing represents a promising on-chip technology that is reproducible, scalable, cost-effective, and a simple method for the production of liposomes and lipid-based nanoparticles [12,32]. Rotational flow and wrapping of fluid streams in microfluidic channels, which permit millisecond mixing at the nanoliter scale, can result in a highly reproducible generation of limited size LNPs (e.g., ≥20 nm). To evaluate the potential of utilizing this approach with SLA archaeosomes, we applied a combination of microfluidics with a design-of-experiment approach to identify an initial set of parameters that would generate material for characterization and in vivo evaluation of their adjuvant activity. Now that we have confirmed the feasibility of generating active SLA archaeosomes with microfluidics, further optimization and analysis could be conducted in future studies using a DoE assisted by computational tools [22]. This would allow us to better understand the effect of each variable as we further develop the microfluidic-based production of SLA archaeosomes. This method has been used for LNP encapsulation of siRNA over a wide range of conditions [11]. The DoE approach used in this study is based on nine different runs with systematic variations in co-solvents, FRR and TFR (Table 1). Utilizing this approach, we identified conditions that allowed us to generate SLA nanoparticles with favorable PDI and zeta potential profiles and average sizes of ~30 or 100 nm (Ethanol/DMSO solvent, with 4:1 FRR/8 mL/min TFR and 1:1 FRR/18 mL/min TFR, respectively). Zeta potential is related to the net electric charge of the particle and the bulk fluid where particles are dispersed. Zeta potential values under different experimental conditions were statistically different from each other, indicating that the choice of co-solvent, FRR and TFR can influence this variable. We noted a solid core appearance in some archaeosomes (Figure 2) that was not previously observed with archaeosomes generated using the thin layer method [10]. Nanoparticles with solid cores have been previously reported in different labs preparing lipid-based nanoaparticles using microfluidic mixing [28]. The formation of solid core archaeosomes can also be investigated using DoE approach in the future. The possible causes of this observation might be because of the formation of different forms of lipid aggregates, micelles or other nanostructures of hydrophobic lipids serving as nucleating sites that could be subsequently coated with other lipid components, leading to bilayer structures [33,34]. It is noteworthy that further optimizations of archaeosomes in terms of biophysical properties and stability could be achieved by modifying experimental conditions. In the present study, we demonstrate that a combined microfluidics DoE approach is a rapid and controllable strategy for the production of archaeosomes of defined sizes. Importantly, the sizes and PDI for archaeosomes of 30 nm and 100 nm remained constant after incubation at 4 °C and 37 °C for three weeks. Based on our previous observation that archaeosomes produced by the thin film hydration method were stable for at least six months at both 4 °C and 37 °C, it is likely that the particles generated by microfluidic-based approaches might also be stable for longer periods of time, but this will need to be confirmed in future studies [21]. Microfluidic mixing represents a suitable technology for producing small and uniform archaeosomes at relatively high concentrations of lipids and using solvents with relatively low toxicity (i.e., ethanol). Compared to thin-film methods, no post-production processing (i.e., sonication, extrusion, freezing and thawing) is required to obtain liposomes of desirable characteristics. In addition, microfluidic mixing, together with automatized downstream purification steps, can be used for high-throughput mass production of liposomes without significantly influencing the quality of the end-product compared to the conventional method. As such, microfluidics-based approached should be considered when looking to upscale production of novel nanoparticle-based therapies/adjuvants such as SLA archaeosomes. Additionally, it is possible to incorporate the vaccine antigen into the microfluidic mixing process with the adjuvant. This will depend on the stability of the antigen when in solution with adjuvant and whether it needs to be stored in lyophilized form. This can be evaluated when SLA archaeosomes are developed with a clinically relevant antigen.

We also demonstrated that, when used as a vaccine adjuvant with OVA, archaeosomes of both 30 nm and 100 nm generated equivalent anti-OVA IgG and cytotoxic CD8+ T cell responses. While the size profile has been shown to be an important factor for activity for other adjuvants/vaccines [15,19,35], it does not seem to be critical for the overall ability of SLA archaeosomes to induce antigen-specific immune responses in the limited size range we have tested. It is possible that much larger or smaller SLA archaeosomes than those tested here could have different adjuvant activity profiles. We did not seek to include much larger particles in our study as they are generally less stable and would be incompatible with sterile filtration methods and difficult to adopt in the future development of SLA as an adjuvant system. We have previously shown that SLA adjuvant formulations produced by thin film hydration appeared to preferentially promote innate immune activation at one day post vaccination, as measured by immune cell recruitment and OVA uptake in the muscle and draining LNs, respectively [36].

As we focused on the more critical ability of the adjuvants to induce antigen-specific immune responses, we have not yet compared the ability of 30 vs. 100 nm SLA archaeosomes to induce the activation of innate immunity, but this could be addressed in future studies. It would also be interesting to evaluate whether SLA could modulate different axes of the immune response involved in allergies, autoimmune disease or pathogenesis of viral infections (e.g., T_reg_, Th17, and IL-31/33) when administered alone or in combination with vitamin D, which has been shown to suppress mast cell activation in allergic manifestations through regulatory T cells [37,38].

Overall, we saw an equivalent biodistribution pattern for both sized archaeosomes and antigen to what we have previously observed with archaeosomes produced using the thin film hydration method [21]. Consistent with our in vivo biodistribution results, an increased retention of OVA at the injection site with SLA archaeosomes of both sizes relative to OVA alone was observed for up to 24 h. Therefore, it would appear the method of archaeosome production, whether by thin film hydration or microfluidics, does not significantly alter archaeosome adjuvanticity or biodistribution, and that at least with the microfluidics method, there was also no significant difference with 30 nm vs. 100 nm archaeosomes. Again, it is possible that greater differences would have been observed with larger archaeosomes. It is worth noting that the correlation between formulation parameters and resultant immune responses is likely complex and may also include additional parameters such as chemical compositions of nanoparticles, nature of the antigens, method of antigen loading, and route of vaccine administration.

## 5. Conclusions

When used as an adjuvant, SLA archaeosomes can induce strong humoral and cell-mediated immune responses to co-administered antigen. While some differences were observed in the antigen retention at the injection site between SLA archaeosome of both sizes produced using microfluidics, overall we have shown that archaeosomes prepared using microfluidic mixing technology with a particle size of 30 nm or 100 nm maintained similar physicochemical properties and adjuvant effects as those generated using traditional thin film hydration methods, yet provide certain advantages such as a better defined particle size, ease of manufacturing and better reproducibility.

## Figures and Tables

**Figure 1 pharmaceutics-14-02291-f001:**
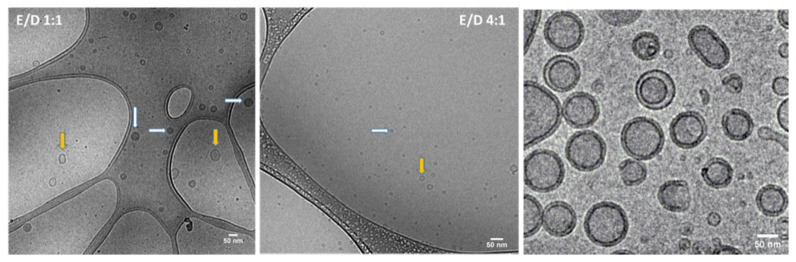
TEM images of SLA-30 and SLA-100 SLA archaeosomes. Nanoparticle morphology was visualized using transmission electron microscopy for SLA-100 (**left**) and SLA30 (**middle**) and those prepared using thin-film hydration (**right**). Two types of vesicles were observed: One displayed a bilayer membrane with hollow aqueous core component (yellow arrow), the other displayed as solid particles with a filled inner compartment (white arrow).

**Figure 2 pharmaceutics-14-02291-f002:**
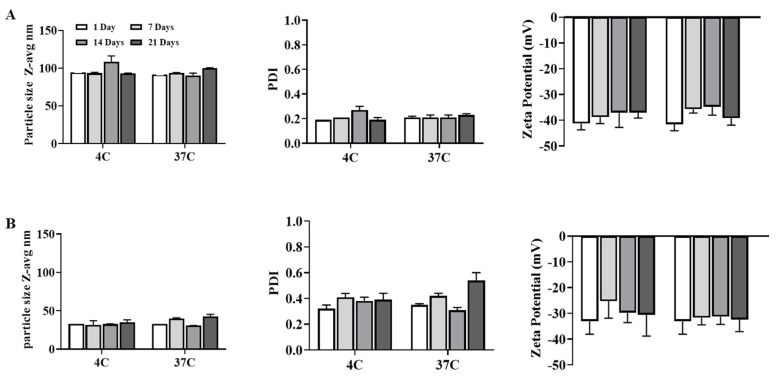
Stability of archaeosomes, 100 nm (**A**); 30 nm (**B**), prepared using microfluidic mixing that were incubated at 4 °C and 37 °C for 3 weeks. All data were measured in the absence of antigen and represent the mean ± SD of three individual measurements.

**Figure 3 pharmaceutics-14-02291-f003:**
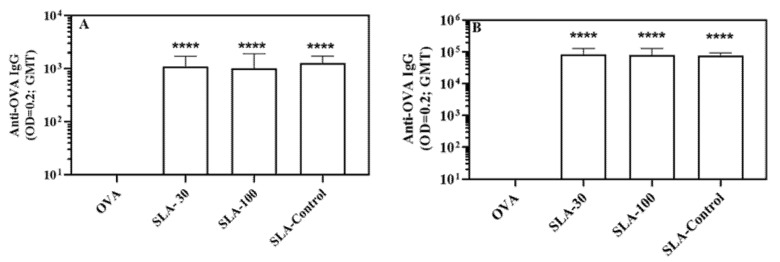
OVA-specific antibody titers in mice. Mice were immunized twice with OVA alone or adjuvanted with SLA archaeosomes at days 0 and 21 by IM. On Day 20 (**A**) and 28 (**B**) (7 days post 2nd immunization) antibody response was assessed by ELISA on individual serum samples. Grouped data is presented as geometric mean + 95% Confidence Interval (n = 4–5 per group). **** *p* < 0.0001.

**Figure 4 pharmaceutics-14-02291-f004:**
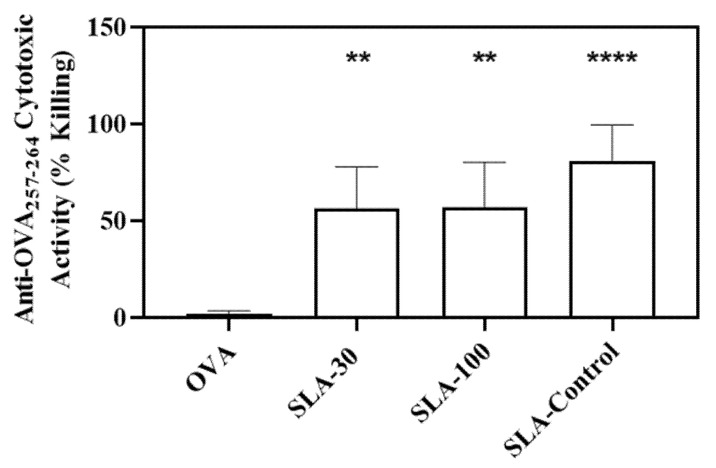
In vivo cytolytic activity in mice. Mice were immunized with OVA antigen alone or adjuvanted with SLA archaeosomes. Target cells (CFSE-labeled splenocytes from naïve mice pulsed with OVA specific CD8 epitope) were transferred to immunized mice on Day 27. On Day 28 (7 days post 2nd immunization), splenocytes were collected and the levels of the target cells measured by flow cytometry. Grouped data is presented as Geometric mean + SD (n = 5 per group). ** *p* < 0.01, **** *p* < 0.0001.

**Figure 5 pharmaceutics-14-02291-f005:**
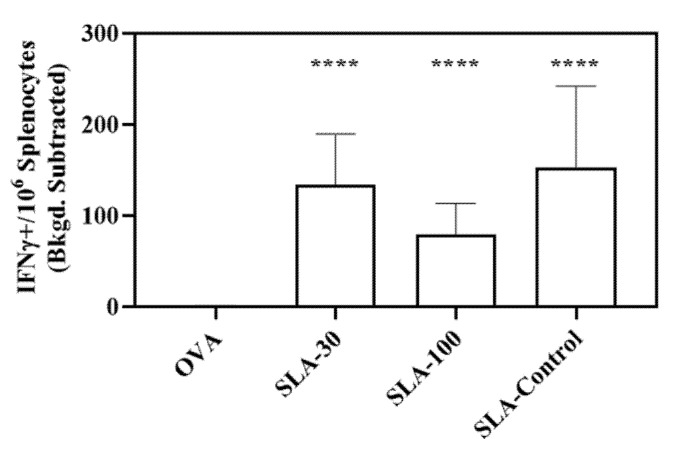
OVA- specific T cells as determined by ELISpot. Splenocytes of mice immunized with OVA antigen alone or formulated with SLA archaeosomes were collected on Day 28 (7 days post 2nd immunization). Cells were stimulated in vitro with OVA peptide SIINFEKL and IFN-γ secretion measured by ELISpot assay. Grouped data is presented as geometric mean ± SD (n = 4–5 per group). **** *p* < 0.0001.

**Figure 6 pharmaceutics-14-02291-f006:**
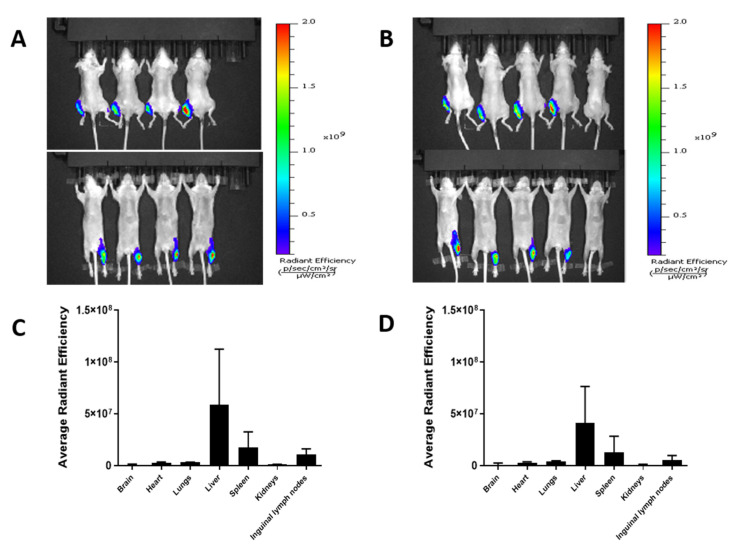
In vivo biodistribution analysis by IVIS imaging of C57BL/6 mice after intramuscular injection of SLA-30 (**A**,**C**) and SLA-100 (**B**,**D**), respectively, at a 24 h time point. (**A**,**B**); whole body images of dorsal (top) and ventral (bottom) of archaeosomes at 24 h postinjection; (**C**,**D**) the averaged fluorescent signals obtained for SLA-30 (**C**) and SLA-100 (**D**), respectively, in major organs at 24 h post injection. Data is presented as Mean ± SD, n = 4/group.

**Figure 7 pharmaceutics-14-02291-f007:**
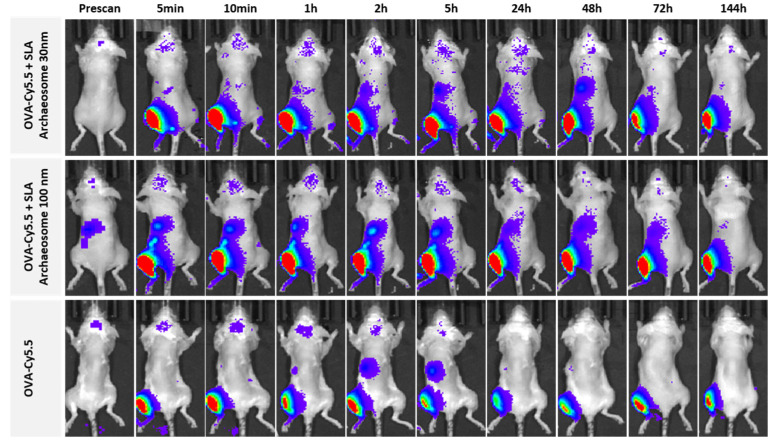
The whole body in vivo biodistribution analysis by IVIS imaging of a representative C57BL/6 mouse on dorsal surface after intramuscular injection of OVA-cy5.5 in the presence of archaeosomes of 30 nm and 100 nm, respectively, and in the absence of archaeosome at predetermined time points.

**Table 1 pharmaceutics-14-02291-t001:** Parameter levels for the main experiment.

Response Parameter/Variable	Level-1	Level-2	Level-3
Co-solvent (*v*:*v*)	E/M (3:1)	E/D (39:1)	E/A (39:1)
FRR (aqueous:lipid; *v*:*v*)	1:1	2:1	4:1
TFR (mL/min)	8	12	18

Note: E: ethanol; A: acetone; M: methanol; D: Dimethyl sulfoxide; FRR: Flow Rate Ratio (aqueous: organic); TFR: Total Flow Rate, mL/min.

**Table 2 pharmaceutics-14-02291-t002:** Design of L_9_(3)^3^ orthogonal array analysis of experimental values. Data are expressed as mean ± SD of three individual measurements (n = 3).

Experimental Runs	Control Parameters			Experimental Values			
	FRR	TFR	Solvent	Z-Avg nm	Percentage of Main Peak	PDI	Zeta Potential mV
1	1:1	8	E/A	94.24 ± 1.95	94.8 ± 3.26	0.27 ± 0.01	−35.73 ± 2.98
2	1:1	12	E/M	112.13 ± 4.45	100.0 ± 0.21	0.23 ± 0.05	−37.83 ± 2.60
3	1:1	18	E/D	94.61 ± 0.29	98.0 ± 2.89	0.19 ± 0.01	−41.27 ± 2.40
4	2:1	8	E/M	153.0 ± 6.58	91.1 ± 5.21	0.51 ± 0.02	−19.83 ± 9.70
5	2:1	12	E/D	346.4 ± 178.74	84.0 ± 10.71	0.68 ± 0.01	−42.03 ± 2.32
6	2:1	18	E/A	59.01 ± 0.96	98.5 ± 1.43	0.39 ± 0.01	−35.6 ± 3.70
7	4:1	8	E/D	32.84 ± 0.04	96.0 ± 2.64	0.29 ± 0.01	−33.07 ± 3.01
8	4:1	12	E/A	30.93 ± 0.95	57.6 ± 12.71	0.64 ± 0.02	−21.64 ± 8.70
9	4:1	18	E/M	212.27 ± 16.9	81.4 ± 5.32	0.40 ± 0.05	−29.43 ± 7.80
		**Thin-film hydration**		94.56 ± 0.36	96.7 ± 2.56	0.27 ± 0.00	−48.00 ±3.48

## Data Availability

The data presented in this study are available on request from the corresponding author. The data are not publicly available due to privacy concerns.

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
