# Peer review of "Evaluation of Adjuvant Activity and Bio-Distribution of Archaeosomes Prepared Using Microfluidic Technology"

_pharmaceutics, 2022, doi:10.3390/pharmaceutics14112291_

Round 1

Reviewer 1 Report

This manuscript presents a further contribution of the authors to the development of adjuvants based on archaeosomes made of the semisynthetic diether lipid SLA. The present contribution is focused on the preparation of archaeosomes by microfluidics; a method that has been explored for liposomes but, until now, not for archaeosomes. The experiments are rather conventional, but the overall organization of the manuscript is appropriate. Results are clearly explained, and conclusions are adequate. I only detected some minor points listed below: 

1.   Authors should explain why they want to prepare 30 nm archaeosomes.

2.   Line 269, Table 2 in place of Table 1

3.   Number of replicates used in orthogonal experimental design should be added

4.   In Figure 1, please explain what is the right image?  

Author Response

thank you for your comments. please see attachment for our responses.

Reviewer 2 Report

The content of a the article is interesting for the importance of the subject matter. Vaccinations are the best baggage for preventing infections that can increase morbidity and mortality. The study improve the knowledgment upon potential adjuvan. The methodology including statystical analysis is adequate and coerent with the study objectives. Figure and tables are adequate. The paper is interesting and well written. I suggest to discuss the impact of adjuvant to specific immune response as Th17 cells and on cytokines release as IL-31/IL-33, and if vitamin D may help the efficacy of vaccine (see and add as references papers by Murdaca et al concerning Th-17 in autoimmune diseases, IL-31/IL--33 axis, vitamin D, Sars-CoV2)

Author Response

(The authors gave the same response as above.)

Reviewer 3 Report

The manuscript describes the preparation and the in vivo assessment of Archaeosomes SLA glycolipid nanoparticles prepared by microfluidic and thin-film methods. Despite the difference in average size, morphology, and preparation methods, both Archeaeospmes showed similar results in terms of adjutants activity, sustained release, and biodistribution. 

The experimental work done in this study provides information needed to move from the traditional method of preparation to a microfluidic-based one. This is interesting for the research project but it is not new and it is already proven that nanoparticles prepared by microfluidic are at least similar to the traditional ones and sometimes better in terms of reproducibility and enhancing the biological activity of associated molecules (doi:10.1088/1361-6528/aa6d15 ). 

The main limitation of this study is that the original approach used (microfluidic and DOE) are partially exploited: 

  • Microfluidic is used to mix an organic solution of lipids and PBS to form liposomes but this is followed by purification steps which are very hard to scale up and could suffer from some reproducibility. so the benefit of microfluidic as a scalable method is not associated with a purification procedure that could be automatized. In addition, no information about the procedure of mixing liposomes with OVA, is this step done by microfluidic? if not, this confirms that the microfluidic approach is not fully exploited.
  • The DOE in this study is used to generate a random set of runs, but the results are not analyzed using software of DOE that could provide relevant and interesting information on the effect of each variable, especially the effect of co-solvent, this information could interesting for Pharmaceutics readers. So the DOE is also partially used and a complete analysis of data could be mandatory to justify the combination of MF and DOE which is interesting because it combines high-quality data provided by microfluidics and advanced analysis of data provided by DOE (doi:10.1021/acsami.2c06627

I would like also to add more specific comments below:  

  • Line 90 to 91: Results are mentioned at the end of the introduction, this should be replaced by highlighting again the main objective of the study (research question) 
  • Line 119, The lipid concentration was 119 based on the manufacturer’s recommendation. the lipids are not locally synthesized? are the recommendation for solubility for microfluidic preparation? 
  • Line 123, error in table number 
  • Line 148, pumping the solvents into the chip is performed by a dual syringe pump. is this correct? usually, chips provided by precision nanosystem are operated with specific apparatus from the same manufacturer, is it possible to have the chips without the full apparatus? 
  • Line 156, the absence of residual solvents is checked by GC, is the lipid concentration in the final samples (after dilution, filtration, … ) is determined by an analytical method or it calculated theoretically? 
  • Line 160, is there any dilution of samples before the measurement?
  • Line 243, how the concentration of particles/mL is calculated?  
  • Line 253, do you use software for DOE analysis? 
  • Line 269 error in table number 
  • Line 272 table 2: some of the samples (runs) indicate a PDI >0.3, is these samples are mono dispersed population, please add in the table a Colonne to indicate the percentage of main peaks (100% for the mono dispersed population, and <100 the polydispersed samples.
  • line 410 DoE approach allows us to rapidly screen and optimizes various archaeology formulations. it is not correct, there is no optimization process because of the partial application of DOE  
  • Line 418 Nanoparticles could not be dissolved? do you mean dispersed? 
  • Line 420, here the difference in morphology should be studied with the methodology to dermine the percentage of each type (solid core or not).
  • Line 471, mentioned that an investigation regarding the influence of other important parameters should be done. my question is whether the data provided by the current study are collected and stored in a proper way to be re-used in future studies that will focus on the other parameters.   

Author Response

thank you for your comments, and please see attachment for our responses.

Round 2

Reviewer 3 Report

Thanks for providing an improved version